# ADAPTIVE GENERATION OF UNRESTRICTED ADVERSARIAL INPUTS

## ABSTRACT

Neural networks are vulnerable to adversarially-constructed perturbations of their inputs. Most research so far has considered perturbations of a fixed magnitude under some $l_p$ norm. Although studying these attacks is valuable, there has been increasing interest in the construction of—and robustness to—*unrestricted* attacks, which are not constrained to a small and rather artificial subset of all possible adversarial inputs. We introduce a novel algorithm for generating such unrestricted adversarial inputs which, unlike prior work, is *adaptive*: it is able to tune its attacks to the classifier being targeted. It also offers a 400–2,000$\times$ speedup over the existing state of the art. We demonstrate our approach by generating unrestricted adversarial inputs that fool classifiers robust to perturbation-based attacks. We also show that, by virtue of being adaptive and unrestricted, our attack is able to defeat adversarial training against it.

## 1    INTRODUCTION

Despite their dramatic successes in other respects, neural networks are well-known to not be adversarially robust. Szegedy et al. (2014) discovered that neural networks are vulnerable to what they termed *adversarial inputs*: by adding carefully-chosen perturbations to correctly-classified inputs, the accuracy of any neural network could be almost arbitrarily decreased. Since then, the machine learning community has rightly focused a great deal of research effort on this phenomenon. Many early efforts to train more robust models initially appeared promising, but have since been shown to be vulnerable to new algorithms for constructing adversarial perturbations (Xu et al., 2019; Athalye et al., 2018). As a result, more attention has been given to methods that provide formal guarantees about performance in the presence of adversarial perturbations (Liu et al., 2019), with the state of the art now providing non-trivial guarantees for the MNIST test set (Wong & Kolter, 2018; Croce et al., 2018; Wang et al., 2018).

However, almost all of this work has focused exclusively on adversarial perturbations whose magnitude is constrained by an $l_p$ norm. There is a growing acknowledgement that this threat model is somewhat contrived: such examples are not a realistic security concern and also occupy a vanishingly small fraction of the set of potential adversarial inputs. Therefore, there is a burgeoning interest in adversarial attacks that are *unrestricted*, in the sense that they do not necessarily derive from a perturbation of a natural input (Brown et al., 2018; Song et al., 2018b).

The main contribution of this paper is a novel and general method to generate unrestricted adversarial inputs. In short, the training procedure for generative adversarial networks (GANs) is modified so that the generator network is rewarded for producing data that are both realistic and deceive a fixed target network. Our approach has four advantages over prior work:

1. Our method is *adaptive* in that it adjusts itself to best attack the specific network being targeted. For instance, adversarial training is ineffective against our approach.

2. Our method is efficient (offering a 400–2000$\times$ speedup over prior work).

3. Our method can easily be applied to *any* existing conditional GAN codebase and checkpoints, regardless of architecture, training procedure, or application domain.

4. Our method therefore demonstrably scales to ImageNet.

## 2 BACKGROUND: GENERATIVE ADVERSARIAL NETWORKS

Generative adversarial networks (GANs) (Goodfellow et al., 2014) are a class of generative machine learning models involving the simultaneous training of two neural networks: a generator $g$ and a discriminator $d$. Specifically, given a dataset $D$ of samples drawn from a probability distribution $p_D$, the generator $g$ learns to transform random noise $z$ drawn from a simple distribution $p_z$ into an approximation of $p_D$. The discriminator network $d$ learns to predict whether a given example $x$ is drawn from the data distribution $p_D$ or was generated by $g$. The generator and the discriminator are *adversarial* because they train simultaneously, with each being rewarded for out-performing the other.

GANs' training behaviours are notoriously temperamental, and many modifications to the original algorithm have been proposed (Goodfellow, 2017). The Wasserstein GAN variant (Arjovsky et al., 2017) aims to provide a more reliable gradient by designing the discriminator (renamed 'critic') to approximate the Wasserstein distance between the distribution generated by $g_\theta$ and the data distribution $p_D$. An additional 'gradient penalty' loss term $L_{gp}$ can be added to implement the constraint that the function be 1-Lipschitz continuous (Gulrajani et al., 2017). The loss functions for this Wasserstein GAN with gradient penalty (WGAN-GP) are: $L_g = \mathbb{E}_{z \sim p_z}[-d(g(z))]$ and $L_d = -L_g + \mathbb{E}_{x \sim p_D}[-d(x)] + \lambda L_{gp}$; where the gradient penalty $L_{gp} = \mathbb{E}_{\tilde{x} \sim p_I}[(\|\nabla_{\tilde{x}} d_\phi(\tilde{x})\|_2 - 1)^2]$, where $p_I$ denotes the distribution sampling uniformly from the linear interpolations between generated samples and examples from $p_D$.

The original proposal for a conditional generative adversarial network (CGAN) learns to generate samples from a conditional distribution (Mirza & Osindero, 2014) by simply passing the intended label $y$ for the generated image to both the generator and the discriminator. An extension of this approach is the auxiliary classifier generative adversarial network (ACGAN) (Odena et al., 2017), in which the discriminator is modified to also predict the label $y$ for the input data. Both the generators are trained to maximise the log-likelihood of the correct label in addition to optimising their usual objective.

## 3 GENERATING UNRESTRICTED ADVERSARIAL INPUTS

Suppose we have a trained target classifier network $f \colon X \to \mathbb{R}^{|Y|}$ that attempts to approximate an oracle function $o \colon O \to Y$ (where $O \subseteq X$ is the oracle's domain) by outputting a confidence $f(x)_c \in \mathbb{R}$ for each class $c \in Y$. As Song et al. (2018b) do, we define an unrestricted adversarial example to be any input $x \in O$ such that the classifier's prediction is incorrect: $\mathrm{argmax}_c f(x)_c \neq o(x)$. Unlike Song et al., we consider the domain of the oracle to be any input with a recognisable class, not just realistic inputs. Gilmer et al. (2018) offer a number motivations for why unrealistic but recongisable adversarial examples could be interesting, including security concerns and improving models' abilities to generalize. Nevertheless, we do carefully evaluate how realistic our results are in Section 4.1.

Unrestricted adversarial examples are a superset of conventional perturbation-based adversarial examples (which are restricted to lie within a fixed distance of some correctly-classified input from a test dataset). While providing a vastly larger space of candidates, a difficulty arises in determining that the classification is incorrect; we can no longer rely on the oracle-provided labels from the test dataset. We leverage generative models to solve this problem.

### 3.1 OUR PROCEDURE

We begin by taking any conditional GAN, with generator loss $l_{ordinary}$; the use of a conditional GAN allows us to determine the correct label $y$ of our generated unrestricted adversarial examples. We then introduce loss terms which incentivise the generator $g$ to create adversarial examples for a classifier $f$. A targeted attack for desired true label $y$ and target label $t \neq y$ should output an image which humans would regard to have label $y$ yet is classified as $t$ by the classifier. We introduce a loss term, $l_{targeted}$, which is minimised when the conditional generator output $g(z,y)$ is classified in this way: $l_{targeted} = \max_{c \neq t} f(g(z,y))_c - f(g(z,y))_t$. For an untargeted attack, we use a loss term that is minimised for *any* misclassification of the example: $l_{untargeted} = f(g(z,y))_y - \max_{c \neq y} f(g(z,y))_c$. Note that these new terms assume that the true labels of the generated data $g(z,y)$ do indeed match the intended labels $y$, an assumption empirically validated in Section 4.1.

Our procedure is to alter the generator's training objective so as to minimise both $l_{ordinary}$ and $l_{(un)targeted}$ simultaneously, thereby training the generator to output data which are both realistic and also fool the target network. Section 3.3 provides further details.

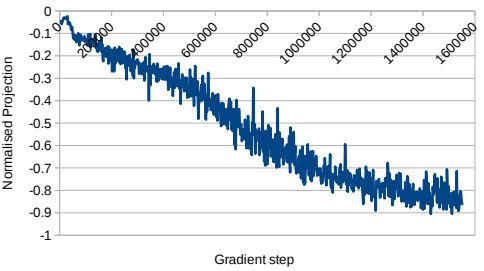
(a) Beginning from a randomly-initialised GAN.

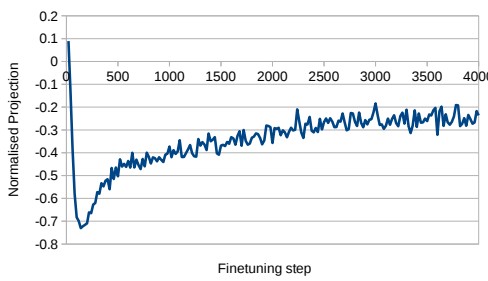
(b) Adversarially finetuning a pretrained GAN.

Figure 1: Projecting normalised gradient vectors from $l_{ordinary}$ and $l_{(un)targeted}$ onto one another.

### 3.2 CHALLENGE: CONFLICTING GRADIENTS

Intuition suggests that the gradient from $l_{ordinary}$ may be pointing in a different direction to the gradient from $l_{(un)targeted}$, since making an image adversarial seems likely to make it *less* realistic, not more. A simple experiment suffices to verify this intuition. We compute the cosine similarity between the gradients of our two loss terms at each step, i.e. $\frac{\nabla l_{ordinary} \cdot \nabla l_{(un)targeted}}{\|\nabla l_{ordinary}\| \|\nabla l_{(un)targeted}\|}$. Figure 1a shows that this projection tends towards $-1$; for reference, if the gradient vectors were selected uniformly at random, the magnitude of this projection would very rarely exceed 0.001. In other words, as training progresses, the gradients from these terms tend towards pointing in *opposite* directions. This makes joint optimisation using a gradient descent approach challenging.

### 3.3 STRATEGIES TO OVERCOME TRAINING CHALLENGES

We empirically evaluate the effect of each technique described below in our ablative experiments reported in Section 4.4.

**Realistic pretraining** It is widely accepted that real image data occupy a relatively low-dimensional and contiguous manifold (Goodfellow et al., 2016, p. 160). Conversely, we know that adversarial examples pervade the full input space: it appears that there is an adversarial example nearby nearly any point in the input space. Therefore, a generator that is pretrained using only $l_{ordinary}$ before *adversarially finetuning* by introducing our additional loss term is more successful than using both loss terms from a random initialisation. By beginning our search in regions of realistic examples, we're more likely to find adversarial examples that are sufficiently realistic. Besides the generated images being subjectively better, Figure 1b shows that the gradients conflict to a much lesser extent. Note that *any* existing conditional GAN architecture, pretrained checkpoint and training algorithm could be used here, allowing our method to leverage the significant advances being made in this area.

**Amalgamation of loss terms** Rather than naïvely summing $l_{ordinary}$ and $l_{(un)targeted}$, we use the following per-example loss term:

$$l_{finetune} = s(l_{ordinary}) \cdot s(l_{(un)targeted} - \kappa), \text{ where } s(l) = \begin{cases} 1 + \exp(l) & \text{if } l \leq 0, \\ 2 + l & \text{otherwise.} \end{cases}$$

Here, $\kappa$ is a hyperparameter similar to that in the Carlini & Wagner (2017) attack: it controls the confidence of the generated adversarial examples. If the difference between the desired logit and the next-greatest logit is less than $\kappa$, the generator is linearly rewarded for improving this gap (gaining confidence); beyond a difference of $\kappa$ (once an example is 'good enough'), the reward exponentially decreases. We use $\kappa = 0$ for our experiments as we do not require strong misclassifications.

**Stochastic loss selection** The gradients from the two loss terms are in conflict, and in practice the $l_{(un)targeted}$ gradient dominates. The proportion of misclassified generated inputs rises quickly to almost 100%, but the generated images were noticeably unrealistic, meaning their correct label may change. To address this, we introduce the 'attack rate' $\mu$. During adversarial finetuning, the finetuning loss term is used at each step only with probability $\mu$; with probability $1 - \mu$, the pretraining loss ($l_{ordinary}$

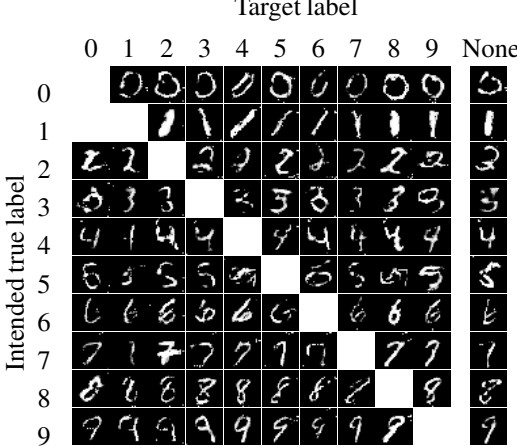

Figure 2: Randomly-selected images generated by a GAN finetuned to attack Wong & Kolter's (2018) classifier, which is robust to perturbations.

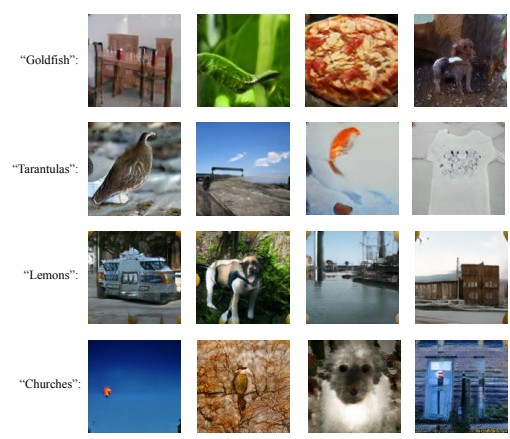

Figure 3: Selected successful targeted unrestricted adversarial examples on ImageNet, generated by a BigGAN (Brock et al., 2019) finetuned to attack ResNet-152 (He et al., 2016).

only) is used. As desired, this new hyperparameter allows the success rate of the generated unrestricted adversarial examples to be traded off with their realism.

## 4 EXPERIMENTAL EVALUATION

Our method aims to generate unrestricted adversarial inputs in a way that adapts to the targeted classifier. We therefore conducted experiments to check whether the generated examples were in fact unrestricted, adversarial, and adapted to the classifier. We then address some questions regarding the performance and generality of our approach, including realism.

The MNIST dataset (LeCun et al., 1998) is the main focus of the experimental evaluation, because this is the most challenging domain for the generation of realistic adversarial inputs. State-of-the-art classifiers perform very well, with around 0.2% test error (Kowsari et al., 2018; Wan et al., 2013). In particular, attempts to create robust classifiers have also been most successful on this dataset, perhaps due to its simplicity (Shafahi et al., 2018). We target five pretrained classifiers provably robust to adversarial perturbations: there is guaranteed to be no adversarial input within a distance $\epsilon$ of $p\%$ of test inputs under the $l_\infty$ norm. All five are the current state-of-the-art in this domain, trained by Wong & Kolter (2018), and Wang et al. (2018). See Appendix C for details.

In our experiments, we combine three well-established generator architectures: a Wasserstein GAN with gradient penalty (WGAN-GP) (Gulrajani et al., 2017), a conditional GAN (Mirza & Osindero, 2014) and an auxiliary classifier GAN (Odena et al., 2017). Full details are given in Appendix E.

A GAN was adversarially finetuned for each of the 10 target labels, and for the untargeted case. Once trained, the generators were used to produce examples for all intended true labels, which were then filtered so that the classifier label matched the target. Images were generated until 200 filtered examples were generated or until 100 seconds had elapsed. Interestingly, this led to no adversarial examples with intended true label '0' and target classification '1', so this case is omitted. Figure 2 and Appendix B give examples of generated images for which the computed label matches the target classification.

### 4.1 EFFICACY OF ATTACKS

We claim that our method generates unrestricted adversarial examples, which are somewhat realistic. We empirically verify each claim in turn.

Since our method does not work by perturbing existing data, only a simple sanity check was required to verify that the generated images are not close to images in the training set, as could be caused by over-fitting. We selected ten generated inputs that are visually similar to the training set, and computed the shortest distances between the images and all images in the training set. The selected images are

Table 1: Comparison of typical perturbation magnitudes from the literature and ours.

| Metric | Nearest neighbour seen | Typical perturbation magnitude |
|---|---|---|
| $l_0$ | 508 | <40 (Ruan et al., 2018) |
| $l_1$ | 22.8 | <5 (Lu et al., 2018) |
| $l_2$ | 3.28 | ~1.5 (Schott et al., 2018) |
| $l_\infty$ | 0.838 | ~0.1 (Wong & Kolter, 2018) |

Table 2: Ten selected unrestricted adversarial inputs used for Table 1.

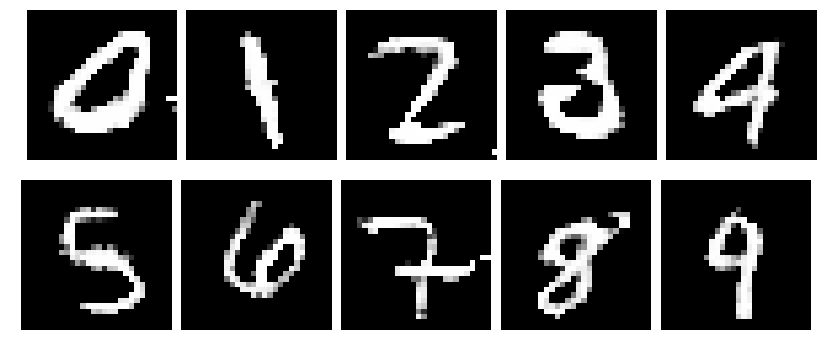

| Intended true label \ Target label | 0 | 1 | 2 | 3 | 4 | 5 | 6 | 7 | 8 | 9 | None |
|---|---|---|---|---|---|---|---|---|---|---|---|
| 0 | | 96 | 94 | 90 | 85 | 96 | 97 | 99 | 85 | 89 | 95 |
| 1 | | | 66 | 88 | 69 | 97 | 89 | 74 | 91 | 81 | 87 |
| 2 | 69 | 89 | | 82 | 58 | 82 | 70 | 64 | 79 | 49 | 75 |
| 3 | 43 | 84 | 81 | | 68 | 74 | 46 | 82 | 54 | 71 | 53 |
| 4 | 84 | 67 | 86 | 74 | | 75 | 96 | 79 | 82 | 77 | 76 |
| 5 | 58 | 75 | 70 | 78 | 79 | | 52 | 82 | 69 | 81 | 75 |
| 6 | 82 | 90 | 95 | 73 | 84 | 84 | | 86 | 94 | 84 | 82 |
| 7 | 75 | 75 | 88 | 82 | 76 | 95 | 88 | | 92 | 59 | 80 |
| 8 | 76 | 85 | 91 | 76 | 98 | 97 | 77 | 75 | | 91 | 83 |
| 9 | 77 | 68 | 90 | 84 | 95 | 92 | 88 | 95 | 95 | | 90 |
| Mean | 70 | 81 | 85 | 81 | 79 | 88 | 78 | 82 | 82 | 76 | 80 |

Figure 4: The success rates of the adversarial attacks by finetuned GANs (the computed label matches the target label and the true label remains the same).

| Intended true label \ Target label | 0 | 1 | 2 | 3 | 4 | 5 | 6 | 7 | 8 | 9 | None |
|---|---|---|---|---|---|---|---|---|---|---|---|
| 0 | | 40 | 60 | 56 | 34 | 46 | 51 | 40 | 36 | 63 | 51 |
| 1 | | | 37 | 52 | 36 | 51 | 81 | 40 | 53 | 35 | 49 |
| 2 | 30 | 37 | | 43 | 40 | 42 | 35 | 37 | 55 | 32 | 54 |
| 3 | 39 | 39 | 43 | | 34 | 40 | 40 | 42 | 45 | 48 | 40 |
| 4 | 51 | 50 | 34 | 38 | | 37 | 46 | 42 | 41 | 43 | 40 |
| 5 | 32 | 34 | 32 | 36 | 43 | | 42 | 36 | 37 | 55 | 51 |
| 6 | 51 | 39 | 45 | 36 | 57 | 46 | | 45 | 57 | 40 | 46 |
| 7 | 47 | 48 | 53 | 33 | 42 | 58 | 41 | | 52 | 44 | 39 |
| 8 | 29 | 46 | 47 | 55 | 44 | 48 | 36 | 39 | | 42 | 60 |
| 9 | 38 | 34 | 50 | 49 | 54 | 53 | 53 | 69 | 57 | | 67 |
| Mean | 40 | 41 | 45 | 44 | 43 | 47 | 47 | 43 | 48 | 45 | 50 |

Figure 5: How often adversarial images are not identified as being generated. If the generated images were completely realistic, the expected result would be 90.

given in Table 2. Table 1 shows that they are much further from any training example than would be the case with a perturbation-based attack.

Next, we evaluate the whether our method is successful in generating adversarial inputs. We need to check that the true label matches the intended true label for each example, or the generator could simply be producing images that visually match the target class. To check this we had Amazon's MTurk workers classify the generated images. For cost reasons, we only did this targeting Wong and Kolter's provably-robust network (Wong & Kolter, 2018). We used a sample size of 100 judges for each intended true label/target label pair for each experiment. Figure 4 shows the proportion of inputs for which not only does the label computed by the classifier match the target label, but the human-judged true label matches the intended true label specified to the generator. The mean number of correct labels for the untargeted attack is 80%. This can be considered to be the success rate of our attack.

We now investigate if the generated examples are realistic. A set of inputs is *realistic* with respect to a dataset if a human cannot reliably identify to which set an example belongs. To check this, we again used MTurk workers. After familiarising themselves with examples from the training dataset, each worker had to pick which image out of ten was most likely to have been generated. Figure 5 shows the proportion of the time that generated images were not identified as such.

For comparison, we repeated these experiments but attacking a non-robust classifier network. The untargeted success rate was 90% (vs. 80% against the robust classifier), and 60% (vs. 50%) were not identified as being generated. Similar differences were seen in targeted attacks; see Appendix I.

### 4.2 Adaptivity to Adversarial Training Defences

In our experiments so far we have evaluated our method against pretrained classifiers that are provably robust to adversarial perturbations. We now investigate whether standard adversarial training (Madry et al., 2018) against our attack in particular is effective. As our method is adaptive, it is not immediately

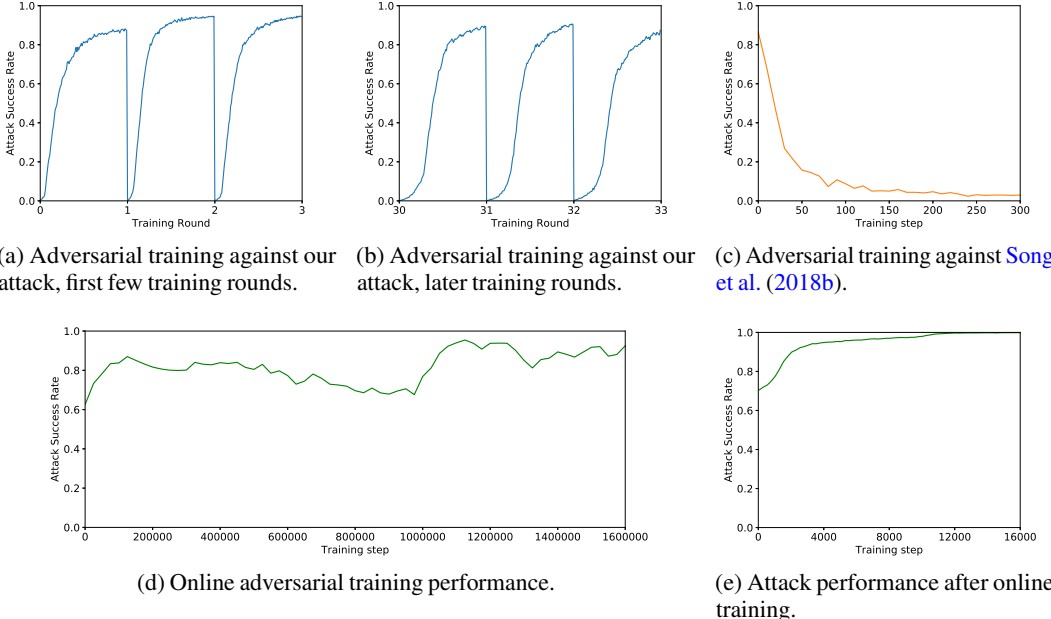

(a) Adversarial training against our attack, first few training rounds.

(b) Adversarial training against our attack, later training rounds.

(c) Adversarial training against Song et al. (2018b).

(d) Online adversarial training performance.

(e) Attack performance after online training.

Figure 6: Plots showing attack efficacy in the presence of adversarial training.

clear how to integrate our method into the adversarial training framework. The two obvious possibilities are finetuning the generator at each step of training, or alternately training the classifier and finetuning the generator. We explore these below as 'online' and 'offline' adversarial training, respectively.

**Online Adversarial Training** We update both the generator and classifier at each training step. We run adversarial training for over 1.6 million training steps, and Figure 6d shows that even during training we maintain a roughly 80% attack success rate. Once we have finished training the classifier, we attack it normally using our method. Despite having been adversarially trained, we are still able to fool the classifier >99% of the time within 16,000 steps, as shown in Figure 6e.

**Offline Adversarial Training** Starting with a pretrained GAN and classifier, we iterate 'training rounds' consisting of two phases. First, a GAN is adversarially finetuned (starting from the pretrained GAN each time) for a fixed period to attack the classifier. Second, 80,000 generated unrestricted adversarial examples are added to the existing training dataset, and the classifier *continues* training until almost 100% accuracy is achieved. Figure 6a shows that, for the first few training rounds, adversarial finetuning is successful: the proportion of examples generated which fool the classifier increases to over 80%. Figure 6b shows the same story 30 rounds (and hence hundreds of thousands of classifier gradient steps) in.

We find that the classifier is able to defend against the kinds of attacks previously produced by the generator. However, the generator's opportunity to adversarially finetune again allows it to generate adversarial examples in a new 'blindspot' of the classifier. For more details on these experiments, see Appendix D.

### 4.3 SCALING TO IMAGENET

To demonstrate the scalability of our method, we apply it to the notoriously large and complex ImageNet-1K dataset, using the author's 'officially unofficial' published code and checkpoints for the current state-of-the-art, BigGAN (Brock et al., 2019). In the untargeted case, our method is able to finetune this BigGAN to fool the classifier >99% of the time within 40 gradient steps (compared to the $10^5$ taken to train from scratch). Our main focus, though, is on the much more challenging targeted attack. We found that typically, on the order of 100 gradient steps were required for >10% of generated examples to be classified (top-1) as the target class. Compared to MNIST, each ImageNet gradient step takes about 100x longer to compute, but the 100x decrease in the number of gradient

steps required compensates for this, resulting in a similar compute time overall. Image quality as measured by Inception Score (Salimans et al., 2016) typically decreased from 70. This is slightly better than mid-2018 state-of-the-art of 52 (Zhang et al., 2019) or the mid-2017 state-of-the-art of 12 using WGAN-GP (Shmelkov et al., 2018). We speculate that if the GAN were finetuned for significantly longer, the gradient from the discriminator would learn to regain some of this lost realism. Figure 3 shows selected samples of generated adversarial examples; Appendix A has a more extensive collection.

### 4.4 ABLATIVE STUDIES AND BASELINE

**No Adversarial Finetuning**   We evaluate the extent to which adversarial finetuning is effective by comparing to a GAN which has not been finetuned. As expected, this is much less successful at producing adversarial examples. The proportion of its outputs which are misclassified matches that of the test set (around 2% for MNIST), whereas adversarial finetuning can easily increase this proportion to well over 99%. Furthermore, only 66% of its misclassified outputs had maintained their true class, as opposed to 80% for an adversarially finetuned GAN. Further results can be found in Appendix K.

**Naïve Adversarial Finetuning**   We investigate the effect of the training strategies described in Section 3.3 by removing each in turn. We find that without pretraining, our method does eventually converge, but with much less realistic results. We also find that the attack rate is an effective technique for trading off adversarial success rate with image realism; without this technique, the attack rate is in effect set to 1, which again produces less realistic results. Note that unrealistic images affect the attack's success rate because the true class of the images is more likely to change. Lastly, we find that choosing $l_{ordinary} + l_{(un)targeted}$ as the generator's loss results in catastrophic collapse of the training, likely because $l_{(un)targeted}$ is optimised for too heavily. Full details can be found in Appendix L.

**Naïve Baseline**   Finally we compare our method to a baseline where data is generated using an ordinary GAN and then adversarially perturbed using projected gradient descent with a norm bound of $\epsilon = 0.1$ under $l_\infty$. With a success rate of under 1% on data which were originally classified correctly, we find this approach is no better than perturbations on the test set; this is as expected, since the generator learns the data distribution.

### 4.5 THREATS TO VALIDITY

The evaluation of the success of the attacks relies on data provided by the MTurk workers. We therefore employed measures to safeguard the quality of this data, described in Appendix H. We also believe that our method will generalise to any dataset and domain for which GANs can be trained successfully. However, this has only been demonstrated on two image classification tasks (albeit dissimilar in nature). Lastly, intuition suggests that our method will be able to adapt to find unrestricted adversarial examples for most fixed defence methods, since it is so free to generate inputs without the constraints that current defence methods rely upon. However, we have only demonstrated it explicitly for the most popular standard defence; we leave it to future work to find a defence against our adaptive approach.

## 5 RELATED WORK

### 5.1 COMPARISON TO THE STATE OF THE ART

We compare our method to that of Song et al. (2018b), the current state of the art in generating unrestricted adversarial examples. Like ours, this method leverages a pretrained GAN. It differs, however, in how adversarial examples are then produced. Instead of adversarially finetuning the generator, it searches for an input to the generator that both deceives the target network and are confidently correctly classified by the discriminator's auxiliary classifier (an ACGAN (Odena et al., 2017) is required in this case). The GAN training is therefore blind to the target network.

Our model achieves similar success rates in generating unrestricted adversarial examples: our success rate of 80% (cf. Section 4.1) is roughly comparable to that of Song et al. (2018b), 88.8%. For comparison, we repeated the realism experiments from Section 4.1, with the difference that judges were asked to identify the one generated image from a choice of two. In this case, Song et al. report that participants select the generated image as the more realistic 21.8% of the time while for our untargeted

attack, this figure is 24%; completely realistic image would be chosen 50% of the time. Full results are given in Appendix J.

Beyond achieving comparable attack success rates, our approach has four significant advantages over prior work. Firstly: adaptivity. In Section 4.2 we have shown that our model is capable of iteratively adapting to an adversarially trained classifier. By contrast, Song et al.'s method performs poorly against adversarial training because the GAN is not trained with respect to a target classifier, remaining fixed after the attack begins. Therefore, if a classifier learns to be correct in the space their algorithm searches, it will no longer be able to generate images different enough to be adversarial. Figure 6c shows that standard adversarial training quickly and effectively defends against Song et al.'s attack, while it fails against ours. Secondly: efficiency. Once trained, our method requires only a single forward pass to generate adversarial examples. Song et al. require 100–500 iterations, each with 4 passes: forward and backward through both the generator and classifier. Our method is therefore 400–2,000× more efficient. Lastly: scale and versatility. Section 4.3 shows that our model scales to ImageNet, a dataset with dimensionality 16× greater than the largest Song et al. demonstrate on. Our method has the further benefit that we can use any pretrained conditional GAN, such as BigGAN (Brock et al., 2019). Song et al. depend on an auxiliary classifier for larger datasets, which BigGAN does not provide.

## 5.2 OTHER RELATED WORK

Wang et al. (2019) independently propose a method which is superficially similar to ours: they also train a GAN to directly generate adversarial examples. Instead of using the ordinary GAN loss to maintain realism, they use a new loss term. This term, $\|g_{pretrained}(z) - g(z)\|_p$, penalises the generator $g$ given input $z$ proportional to the deviation caused by finetuning from the original output. Wang et al.'s choice of loss term has the unfortunate effect of preventing the generator from generating either unrestricted adversarial examples or examples which are sure to fall within an $l_p$-norm ball of a realistic input. By contrast our approach allows for truly unrestricted and adaptive examples. Furthermore: we evaluate against state-of-the-art provably-robust networks rather than ad-hoc classifiers; we do not assume that all the true labels remain the same (which is unlikely), and conduct a user study to test this; and we demonstrate that our approach scales beyond MNIST (to ImageNet).

Sharif et al. (2019) train a network to generate patterned spectacles, which, when added to an image of a face, cause misclassification. They also adapt this approach to MNIST using an approach quite similar to ours. However, this only achieves a success rate of 0.83% after filtering to "only the digits that where likely to be comprehensible by humans" against a classifier which was state-of-the-art in 2017. In contrast, we achieve around 80% accuracy against current state-of-the-art robust classifiers.

A wide range of work trains networks to generate adversarial *perturbations* (Hayes & Danezis, 2018; Baluja & Fischer, 2018; Xiao et al., 2018; Song et al., 2018a; Poursaeed et al., 2018). While these must also balance realism and adversarial success, the key difference is that we generate *unrestricted* adversarial examples, allowing attacks to succeed when constrained perturbations provably fail.

Hu et al. (2019) introduce a search for pairs of nearby unrestricted adversarial examples, but unfortunately cannot ensure that their true label is meaningful; if the search starting point is random, it is overwhelmingly likely not to be. If instead it is a known input, the examples are not unrestricted.

## 6 CONCLUSION

We have introduced an algorithm which trains a GAN to generate unrestricted adversarial inputs; we demonstrate that these, as expected, are successful against state-of-the-art classifiers robust to perturbation attacks. The key novelty in our attack procedure is that it entails the tuning of the weights of the generator to target a specific network. As a result, it can be considered *adaptive*: we have shown that, while prior work is quickly mitigated by standard adversarial training, our attack adapts to find a new way of fooling the classifier. In addition, once the generator is adversarially finetuned, it becomes an endless supply of cheap adversarial examples: generation of adversarial examples requires a single forward pass rather than execution of any optimisation algorithm, resulting in a 400–2000× speedup over the state of the art. We have also demonstrated that any existing GAN codebase can easily be used by adapting BigGAN to generate unrestricted adversarial examples for ImageNet.

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

## A    SAMPLES OF IMAGENET UNRESTRICTED ADVERSARIAL EXAMPLES

Randomly-selected successful targeted unrestricted adversarial examples generated using adversarially finetuned BigGANs (Brock et al., 2019). The targeted classifier is ResNet-152 (He et al., 2016), the highest-accuracy pretrained classifier packaged with PyTorch. Besides setting our attack rate at 0.1, all configuration and hyperparameters are as described in the BigGAN 'officially unofficial' codebase.[1]

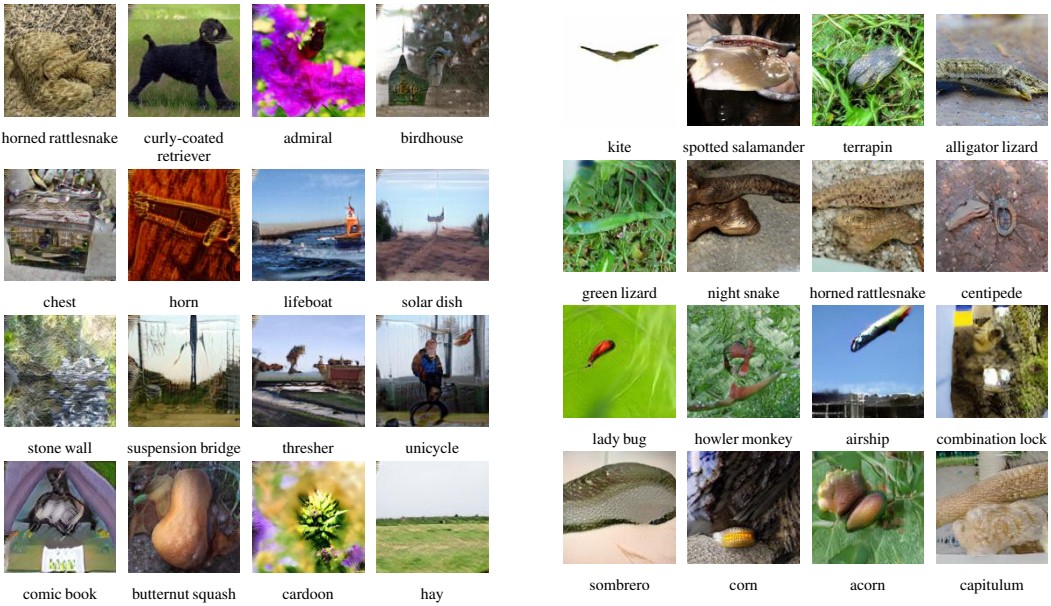

Figure 7: Successful targeted unrestricted adversarial examples for target class 'tabby cat'.

Figure 8: Successful targeted unrestricted adversarial examples for target class 'slug'.

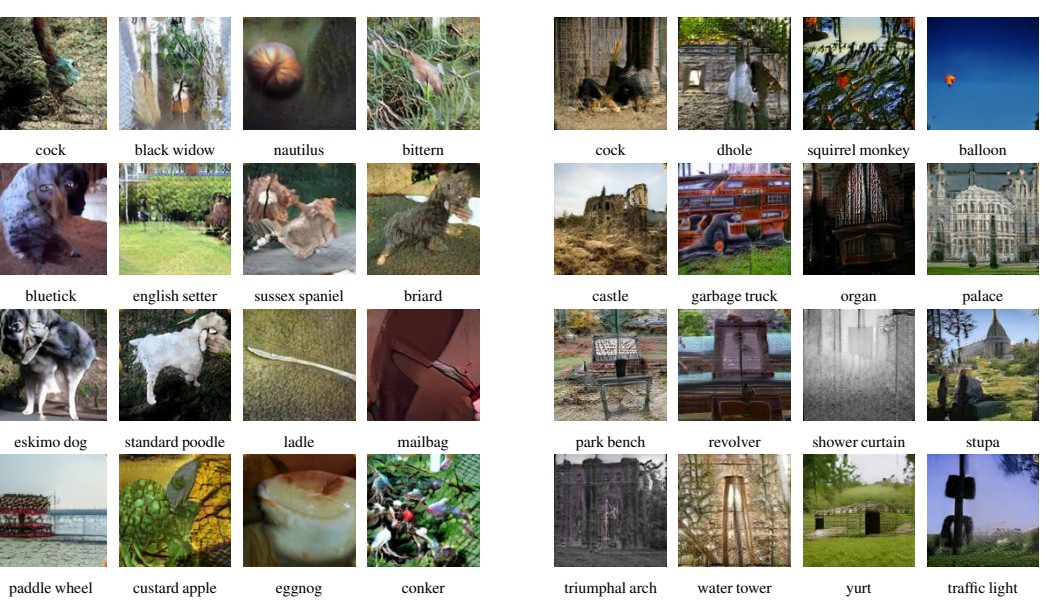

Figure 9: Successful targeted unrestricted adversarial examples for target class 'orange'.

Figure 10: Successful targeted unrestricted adversarial examples for target class 'church'.

[1]https://github.com/ajbrock/BigGAN-PyTorch

## A.1 Comparison to No Adversarial Finetuning

The results above are not as high-quality as those reported in the BigGAN paper (Brock et al., 2019). There are two causes of this which need to be disentangled: the effect of adversarial finetuning, and the limitations of the BigGAN implementation. One limitation is that the checkpoint we use has a much lower Inception Score than is reported in the paper. Another is that our limited access to expensive hardware forces us to use a batch size of 15; small batch sizes "lead to inaccurate estimation of the batch statistics, and reducing batch normalisation's batch size increases the model error dramatically" (Wu & He, 2018, p.1).

We therefore provide samples from the BigGAN running on our machine, 15 iterations after the checkpoint. Figure 11a shows a set of samples taken after 15 iterations of continuing ordinary training; Figure 11b shows the output of the generator for the same input after 15 iterations of adversarial finetuning, instead.

To our eyes, the samples in Figure 11a are similar in quality to those in Figure 11b and on the previous page.

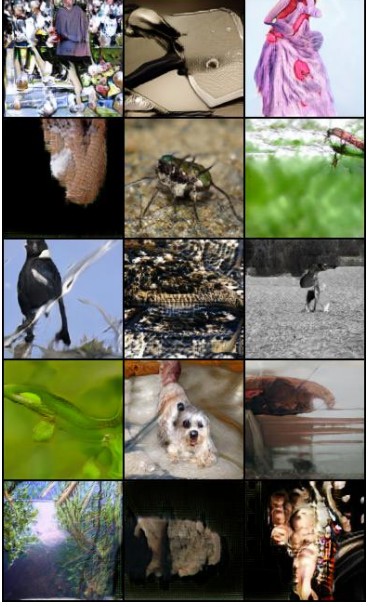

(a) No adversarial finetuning: ordinary training continues.

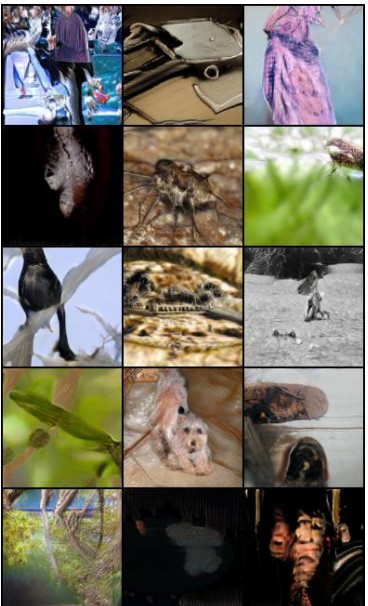

(b) With 15 gradient steps of adversarial finetuning.

Figure 11: Samples for fixed inputs to the BigGAN implementation we use, taken 15 gradient steps after the checkpoint we begin adversarial finetuning from.

## B  SAMPLES OF MNIST UNRESTRICTED ADVERSARIAL EXAMPLES

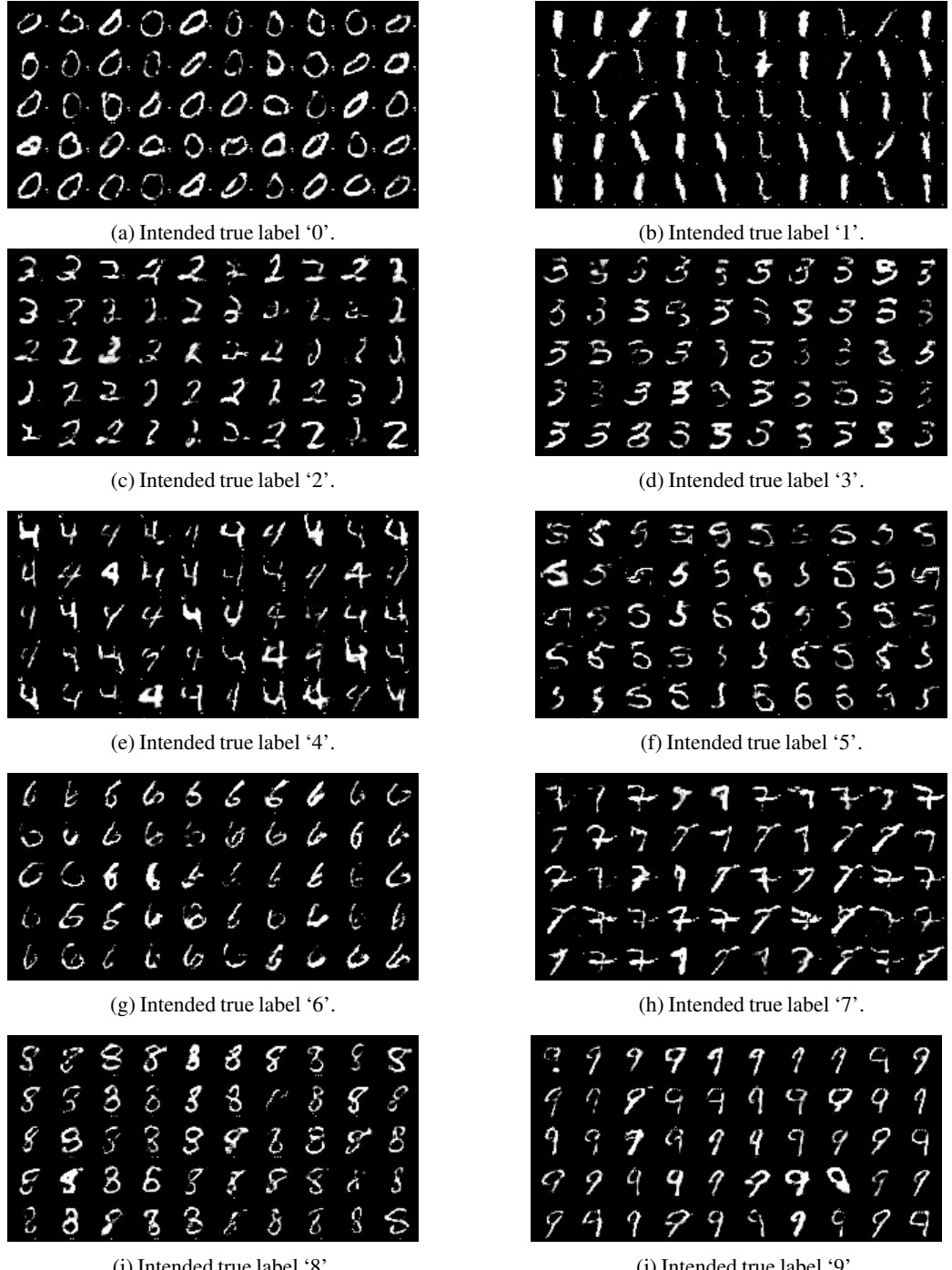

(a) Intended true label '0'.                    (b) Intended true label '1'.

(c) Intended true label '2'.                    (d) Intended true label '3'.

(e) Intended true label '4'.                    (f) Intended true label '5'.

(g) Intended true label '6'.                    (h) Intended true label '7'.

(i) Intended true label '8'.                    (j) Intended true label '9'.

Figure 12: Examples generated by one adversarially-finetuned GAN to perform an untargeted attack on Wong & Kolter's (2018) classifier, which is provably robust to perturbation attacks.

## C  TARGETED CLASSIFIERS

All targeted classifiers (other than 'simple fully-connected') are provably robust to adversarial perturbations in the sense that there is guaranteed to be no adversarial input within a distance $\epsilon$ of $p\%$ of test inputs under the $l_\infty$ norm.

Table 3: Descriptions of and references to target classifiers used.

| Our Name | Abbreviation | $\epsilon$ | $p$ | Architecture |
|---|---|---|---|---|
| Wong & Kolter (2018) | W&K | 0.1 | 94.2 | 2 convolutional layers followed by 2 dense layers |
| MixTrain (Wang et al., 2018) Model A | MT-A | 0.1 | 97.1 | 'MNIST_small': 2 convolutional layers followed by 1 dense layer |
| MixTrain (Wang et al., 2018) Model B | MT-B | 0.3 | 60.1 | 'MNIST_small': 2 convolutional layers followed by 1 dense layer |
| MixTrain (Wang et al., 2018) Model C | MT-C | 0.1 | 96.4 | 'MNIST_large': 4 convolutional layers followed by 2 dense layers |
| MixTrain (Wang et al., 2018) Model D | MT-D | 0.3 | 58.4 | 'MNIST_large': 4 convolutional layers followed by 2 dense layers |
| Simple Fully-Connected | Simple | N/A | N/A | Three fully-connected layers of size 256, 128 and 32 with LeakyReLU activations |

## D  ADVERSARIAL TRAINING EXPERIMENT

The classifier trained during adversarial training (both the architecture and hyperparameters) is the one used in Madry et al. (2017), and in particular from their associated MNIST Adversarial Examples Challenge.

For the offline experiments with our own model, we first pretrain the generator. We then continue in 'training rounds'. First, we fine-tune against the classifier for 5000 gradient steps, using the hyperparameters from Table 6, but with an attack rate of 0.4. Next, we produce 80,000 attacked training examples (using an untargeted attack), which are added to the pool of all examples generated so far. Then, the classifier is trained on the entirety of the pool of samples 30 times, with a batch size of 128. Once a training round is completed we start again, resetting the GAN to how it was *before* the adversarial finetuning.

For the experiments with Song et al.'s (2018b) model, we run 300 training gradient steps for the Madry et al. classifier, with a batch size of 64. At each step, the training data is produced by Song et al.'s model. We use their code and the hyperparameters they provide for untargeted attacks in Table 4 of their appendix.

# E  MNIST EXPERIMENTS: ARCHITECTURES AND HYPERPARAMETERS

The WGAN-GP (Gulrajani et al., 2017) and ACGAN (Odena et al., 2017) architectures were the starting points for the design of these neural networks. Only a small amount of manual hyperparameter tuning was performed.

The generator is a convolutional neural network, conditioned on class label.

The discriminator network is a combination of a conditional WGAN-GP critic, which learns an approximation of the Wasserstein distance between the generated and training-set conditional distributions, and an auxiliary classifier, which predicts the likelihood of the possible values of $h(x)$. We combined these two architectures in an attempt to strengthen the gradient provided to the generator, helping to generate data which are both realistic and for which the true (i.e., human-judged) labels match the intended true labels. The critic is given the true label of the data $h(x)$ to improve its training, but the auxiliary classifier must not have access to this information since its purpose is to predict it. We therefore split the discriminator $d$ into three sub-networks. Network $d_0 \colon X \to \mathbb{R}^i$ effectively preprocesses the input, passing an intermediate representation to the critic network $d_1 \colon \mathbb{R}^i \times Y \to \mathbb{R}$ and the auxiliary classifier network $d_2 \colon \mathbb{R}^i \to \mathbb{R}^{|Y|}$. In our experiments, both $d_1$ and $d_2$ were single fully-connected layers of the appropriate dimension. The loss terms from the WGAN-GP and ACGAN algorithms are simply summed. The auxiliary classifier helps the training converge, but is not necessary.

Table 4: Architecture for generator network, $g$.

| Layer Type | Kernel | Strides | Feature Maps | Batch Norm. | Dropout | Activation |
|---|---|---|---|---|---|---|
| Fully-Connected | N/A | N/A | | No | 0 | ReLU |
| Transposed Convolution | $5\times5$ | $2\times2$ | 64 | Yes | 0.35 | LeakyReLU |
| Transposed Convolution | $5\times5$ | $2\times2$ | 32 | Yes | 0.35 | LeakyReLU |
| Transposed Convolution | $5\times5$ | $2\times2$ | 8 | Yes | 0.35 | LeakyReLU |
| Fully-Connected | N/A | N/A | 4  784 | No | 0 | Tanh |

Table 5: Architecture for discriminator subnetwork, $d_0$.

| Layer Type | Kernel | Strides | Feature Maps | Batch Norm. | Dropout | Activation Function |
|---|---|---|---|---|---|---|
| Convolution | $3\times3$ | $2\times2$ | 8 | No | 0.2 | LeakyReLU |
| Convolution | $3\times3$ | $1\times1$ | 16 | No | 0.2 | LeakyReLU |
| Convolution | $3\times3$ | $2\times2$ | 32 | No | 0.2 | LeakyReLU |
| Convolution | $3\times3$ | $1\times1$ | 64 | No | 0.2 | LeakyReLU |
| Convolution | $3\times3$ | $2\times2$ | 128 | No | 0.2 | LeakyReLU |
| Convolution | $3\times3$ | $1\times1$ | 256 | No | 0.2 | LeakyReLU |

Table 6: Hyperparameters for all networks.

| Hyperparameter | Value |
|---|---|
| Attack rate | $\mu = 0.1$ |
| Learning rate | $\alpha = 0.000005$ |
| Adam betas | $\beta_1 = 0.6, \beta_2 = 0.999$ |
| Leaky ReLU slope | 0.2 |
| Minibatch size | 100 |
| Dimensionality of latent space | 128 |
| Weight initialisation | Normally distributed as described by He et al. (2015) |
| Coefficient of gradient penalty loss term | $\lambda = 10$ |

## F    VISUAL EFFECT OF ADVERSARIAL FINETUNING

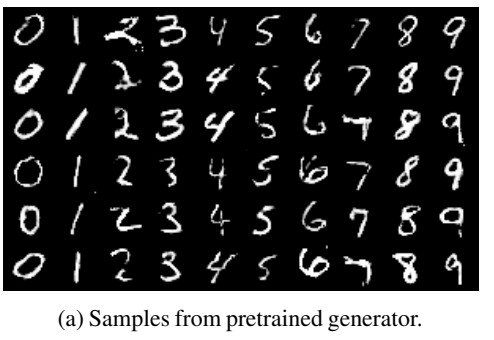

(a) Samples from pretrained generator.

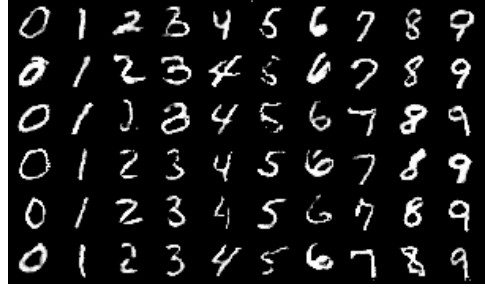

(b) After 5,000 iterations of finetuning.

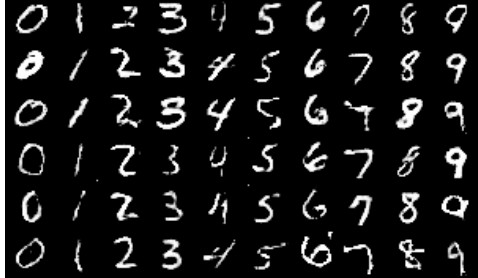

(c) After 10,000 iterations of finetuning.

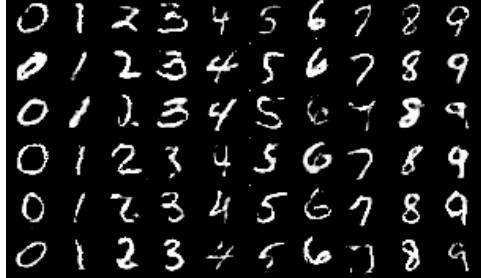

(d) After 20,000 iterations of finetuning.

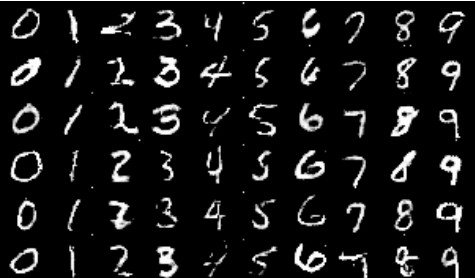

(e) After 30,000 iterations of finetuning.

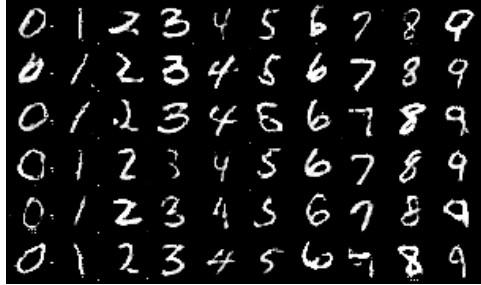

(f) After 45,000 iterations of finetuning. We ended finetuning at this stage.

Figure 13: A sequence of images tracking the output of the generator network for one fixed random sample in latent space as adversarial finetuning takes place. Five samples are given for each intended true label. The finetuning is an untargeted attack against Wong & Kolter's (2018) provably-robust network.

## G    TRANSFERABILITY OF ADVERSARIAL EXAMPLES

Perturbation-based adversarial examples typically somewhat generalise between models (Szegedy et al., 2014; Liu et al., 2017). That is, inputs crafted using white-box access to fool one model often fool a different model. This means that black-box attacks are possible, if the attacker has a different trained model for the same task. To evaluate whether our method could be used in the same way, we generated about 20,000 untargeted unrestricted adversarial inputs for each target classifier, and measured the misclassification rates on this set for the other models. The high variance of the results, shown in Table 7, suggests that successful transfer may depend more on the networks in question than on our generation algorithm.

Table 7: The percentage of adversarial examples targeting each classifier which are also adversarial for the others. See Appendix C for descriptions of the classifiers.

|  |  | To | | | | | |
|---|---|---|---|---|---|---|---|
|  |  | W&K | MT-A | MT-B | MT-C | MT-D | Simple |
| From | W&K |  | 20.2 | 18.4 | 9.0 | 60.7 | 16.8 |
|  | MT-A | 19.5 |  | 14.1 | 13.3 | 55.2 | 4.7 |
|  | MT-B | 5.2 | 4.8 |  | 1.6 | 57.8 | 2.6 |
|  | MT-C | 25.8 | 47.6 | 13.9 |  | 67.8 | 12.1 |
|  | MT-D | 5.9 | 7.3 | 9.4 | 4.3 |  | 1.7 |
|  | Simple | 2.7 | 2.6 | 2.6 | 1.3 | 48.0 |  |

## H    SAFEGUARDING MTURK DATA QUALITY

The evaluation of our method relies entirely on the quality of the data provided by the MTurk workers. We therefore took a number of measures to ensure that participants understood the instructions and completed the tasks diligently:

- Only workers with good track records were permitted to participate.
- The instructions specified that particular answers should be given to specified questions to prove that the instructions had been read carefully. Approximately 10% of work was rejected for failing this check.
- For the image labelling tasks, some images with known labels were included to check that the right labels were being given. Reassuringly, almost no work was rejected for failing this check.
- For the identification of the generated images, a bonus nearly doubling the pay per image was given for each correctly-identified image, providing an extra incentive to try hard.
- To provide a disincentive to high-speed random clicking, a minimum time spent answering each question was enforced.
- If more than 1% of questions were left unanswered, we interpreted this as a sign of carelessness and did not use any of the data from that task.

# I RESULTS FOR NON-ROBUST TARGET NETWORK

These results are targeting a simple convolutional neural network with LeakyReLU activations and three hidden layers of size 256, 128 and 32, trained until convergence.

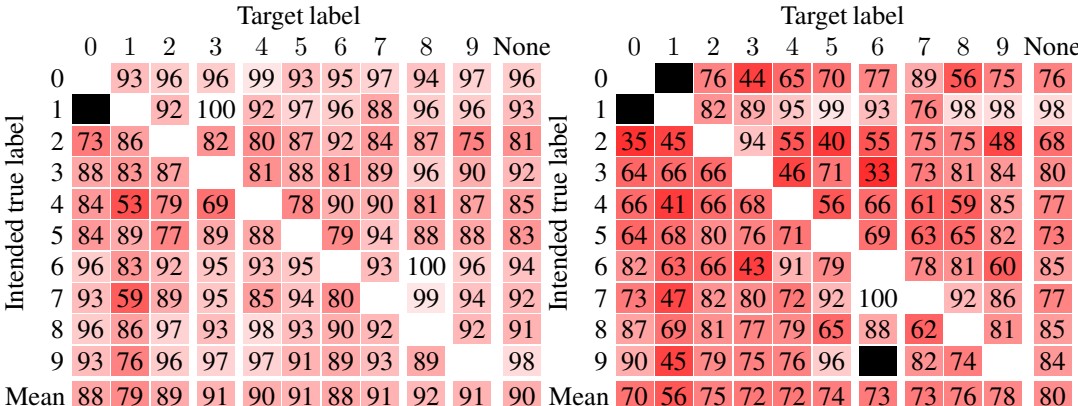

Figure 14: The success rates of the adversarial attacks by finetuned GANs. More precisely, of generated images for which the computed label output by the classifier matches the target label, the percentage which are truly adversarial (in the sense that the true label of the image matches the intended true label passed to the generator network) is reported.

Figure 15: The success rates of the adversarial attacks by a pretrained but not finetuned GAN. More precisely, of generated images for which the computed label output by the classifier matches the target label, the percentage which are truly adversarial (in the sense that the true label of the image matches the intended true label passed to the generator network) is reported.

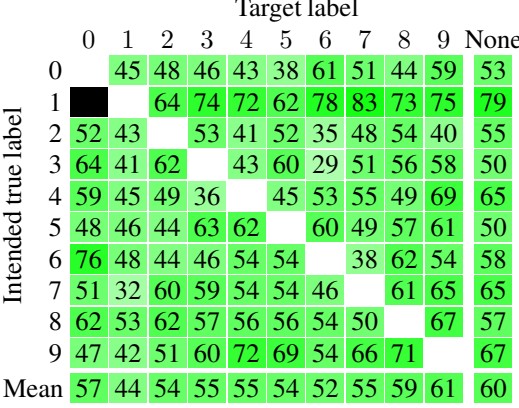
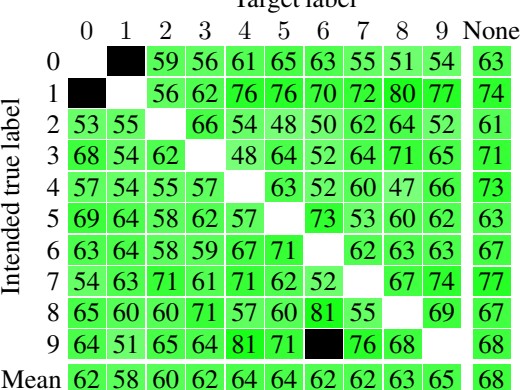

Figure 16: Measures of how realistic the adversarial images generated by finetuned GANs are. More precisely, the proportion of generated inputs for which the classified label matches the target label which were not identified as being generated when placed amongst nine images from the training dataset. If the generated images were completely realistic, the expected result would be 90.

Figure 17: Measures of how realistic the adversarial images generated by a pretrained but not finetuned GAN are. More precisely, the proportion of generated inputs for which the classified label matches the target label which were not identified as being generated when placed amongst nine images from the training dataset. If the generated images were completely realistic, the expected result would be 90.

## J    Side-by-Side Image Comparison Results

Each figure shows the number of human judgements out of 100 which correctly identified the unrestricted adversarial input in a side-by-side comparison with an image drawn from the dataset. If the generated images were completely realistic, the expected result would be 50.

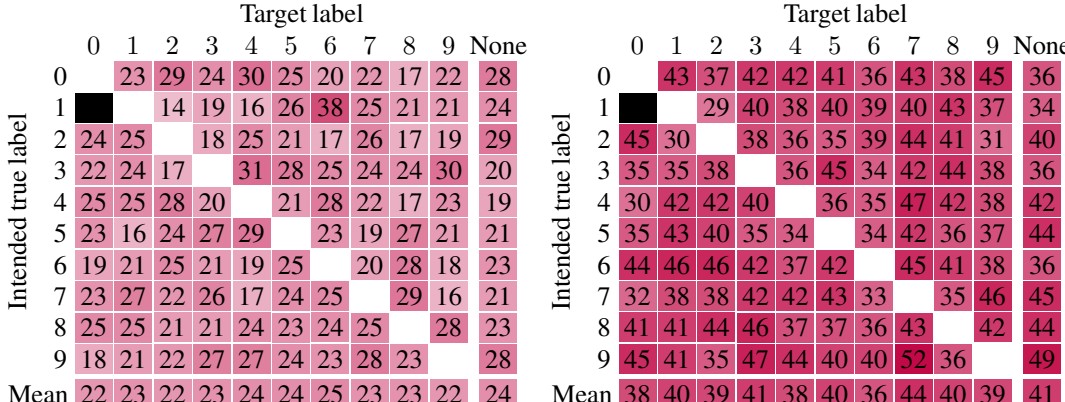

Figure 18: Results against Wong & Kolter (2018) generated by adversarially finetuned GANs.

Figure 19: Results against Wong & Kolter (2018) generated by a pretrained but not finetuned GAN.

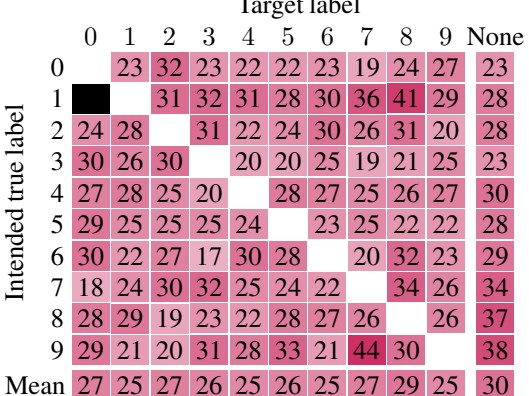
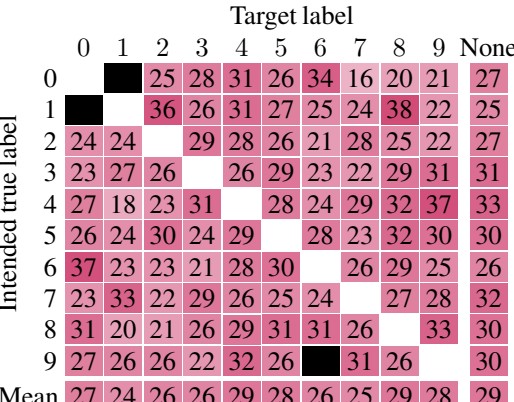

Figure 20: Results against an ordinary neural network generated by adversarially finetuned GANs.

Figure 21: Results against an ordinary neural network generated by a pretrained but not finetuned GAN.

## K ADVERSARIAL FINETUNING ABLATION STUDY

We perform the attack procedure described in Section 4, but using a GAN which has *not* been adversarially finetuned. That is, we use the generator to generate many examples, and filter to keep all those which are misclassified (the untargeted case) or misclassified with a particular label (the 'targeted' case, although note that of course the un-finetuned generator has no notion of a target). We then evaluate the proportion of these filtered examples which correctly maintain their intended true class, a necessary condition for an adversarial attack. We also report the proportion of these filtered examples which are not correctly identified by humans judges as being generated (out of a selection of ten).

For instance, consider the result for intended true label 9 and target label 0. We first use the conditional GAN to produce a set of images that are intended to be 9s. We then filter this set and keep only those that are classified as 0s by the classifier. Finally, we report below the percentage of these for which the true label (determined by humans) is indeed a 9 (55%), and the percentage of these which fool human judges into believing that they are real test data (51%).

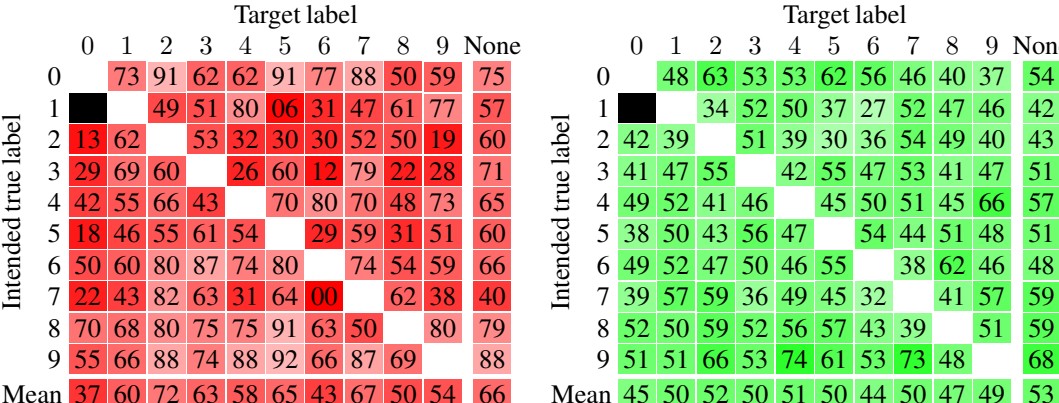

Figure 22 (left): Adversarial success rates for a non-finetuned generator.

| Intended true label \ Target label | 0 | 1 | 2 | 3 | 4 | 5 | 6 | 7 | 8 | 9 | None |
|---|---|---|---|---|---|---|---|---|---|---|---|
| 0 |  | 73 | 91 | 62 | 62 | 91 | 77 | 88 | 50 | 59 | 75 |
| 1 |  |  | 49 | 51 | 80 | 06 | 31 | 47 | 61 | 77 | 57 |
| 2 | 13 | 62 |  | 53 | 32 | 30 | 30 | 52 | 50 | 19 | 60 |
| 3 | 29 | 69 | 60 |  | 26 | 60 | 12 | 79 | 22 | 28 | 71 |
| 4 | 42 | 55 | 66 | 43 |  | 70 | 80 | 70 | 48 | 73 | 65 |
| 5 | 18 | 46 | 55 | 61 | 54 |  | 29 | 59 | 31 | 51 | 60 |
| 6 | 50 | 60 | 80 | 87 | 74 | 80 |  | 74 | 54 | 59 | 66 |
| 7 | 22 | 43 | 82 | 63 | 31 | 64 | 00 |  | 62 | 38 | 40 |
| 8 | 70 | 68 | 80 | 75 | 75 | 91 | 63 | 50 |  | 80 | 79 |
| 9 | 55 | 66 | 88 | 74 | 88 | 92 | 66 | 87 | 69 |  | 88 |
| Mean | 37 | 60 | 72 | 63 | 58 | 65 | 43 | 67 | 50 | 54 | 66 |

Figure 23 (right): Realism rates for adversarial images produced by a non-finetuned generator.

| Intended true label \ Target label | 0 | 1 | 2 | 3 | 4 | 5 | 6 | 7 | 8 | 9 | None |
|---|---|---|---|---|---|---|---|---|---|---|---|
| 0 |  | 48 | 63 | 53 | 53 | 62 | 56 | 46 | 40 | 37 | 54 |
| 1 |  |  | 34 | 52 | 50 | 37 | 27 | 52 | 47 | 46 | 42 |
| 2 | 42 | 39 |  | 51 | 39 | 30 | 36 | 54 | 49 | 40 | 43 |
| 3 | 41 | 47 | 55 |  | 42 | 55 | 47 | 53 | 41 | 47 | 51 |
| 4 | 49 | 52 | 41 | 46 |  | 45 | 50 | 51 | 45 | 66 | 57 |
| 5 | 38 | 50 | 43 | 56 | 47 |  | 54 | 44 | 51 | 48 | 51 |
| 6 | 49 | 52 | 47 | 50 | 46 | 55 |  | 38 | 62 | 46 | 48 |
| 7 | 39 | 57 | 59 | 36 | 49 | 45 | 32 |  | 41 | 57 | 59 |
| 8 | 52 | 50 | 59 | 52 | 56 | 57 | 43 | 39 |  | 51 | 59 |
| 9 | 51 | 51 | 66 | 53 | 74 | 61 | 53 | 73 | 48 |  | 68 |
| Mean | 45 | 50 | 52 | 50 | 51 | 50 | 44 | 50 | 47 | 49 | 53 |

Figure 22: Adversarial success rates for a non-finetuned generator, as described above.

Figure 23: Realism rates for adversarial images produced by a non-finetuned generator, as described above. If the generated images were completely realistic, the expected result would be 90.

## L    TRAINING STRATEGIES ABLATION STUDIES

We run several ablation studies for the components described in Section 3.3. We find that while these strategies are not necessary, they make examples more realistic. This in turn improves the chance that the true class of an example remains the same, increasing our success rate.

### L.1    PRETRAINING

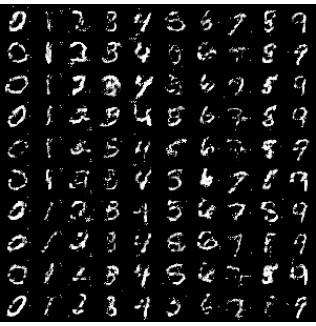
(a) Results after having run as long as pretraining, 705,000 iterations

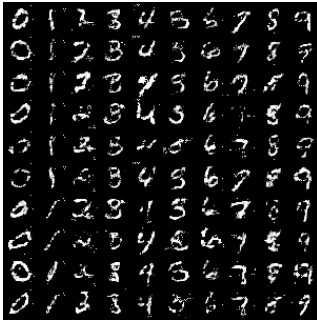
(b) Results after having run as long as Figure 13, 750,000 iterations

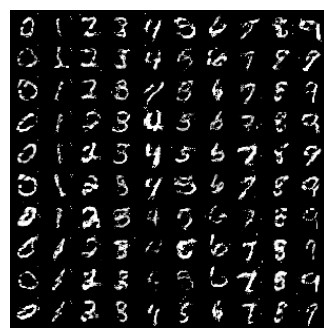
(c) Results once training had converged, over 2.5 million iterations

Figure 24: On MNIST we pretrained the GAN for 705,000 iterations, and then finetuned for another 45,000. Here, we show results using the same setup as in Figure 13. We show the results after the number of iterations equivalent to pretraining, to pretraining and finetuning, and at convergence. The final results both took longer and are visually less convincing than comparable results in Figure 13.

### L.2    NAÏVE LOSS FUNCTION

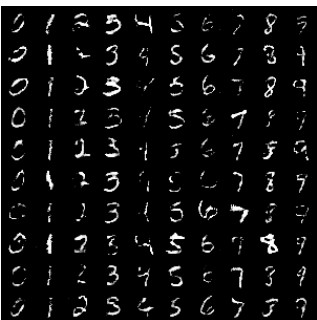
(a) After 500 adversarial finetuning steps; 40% misclassified.

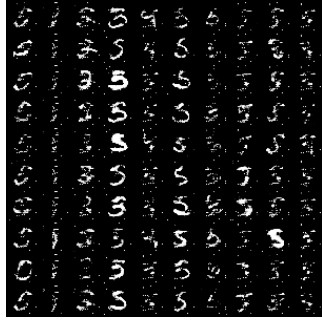
(b) After 1500 adversarial finetuning steps; 91% 'misclassified'.


(c) After 10,000 adversarial finetuning steps; all 'misclassified'.

Figure 25: Adversarial finetuning with naïve generator loss $l_{ordinary} + l_{untargeted}$. As expected, our custom loss function as described in Section 3.3 significantly improves convergence to generator weights which continue to generate images realistic enough to maintain their true classes.

## L.3    ATTACK RATE

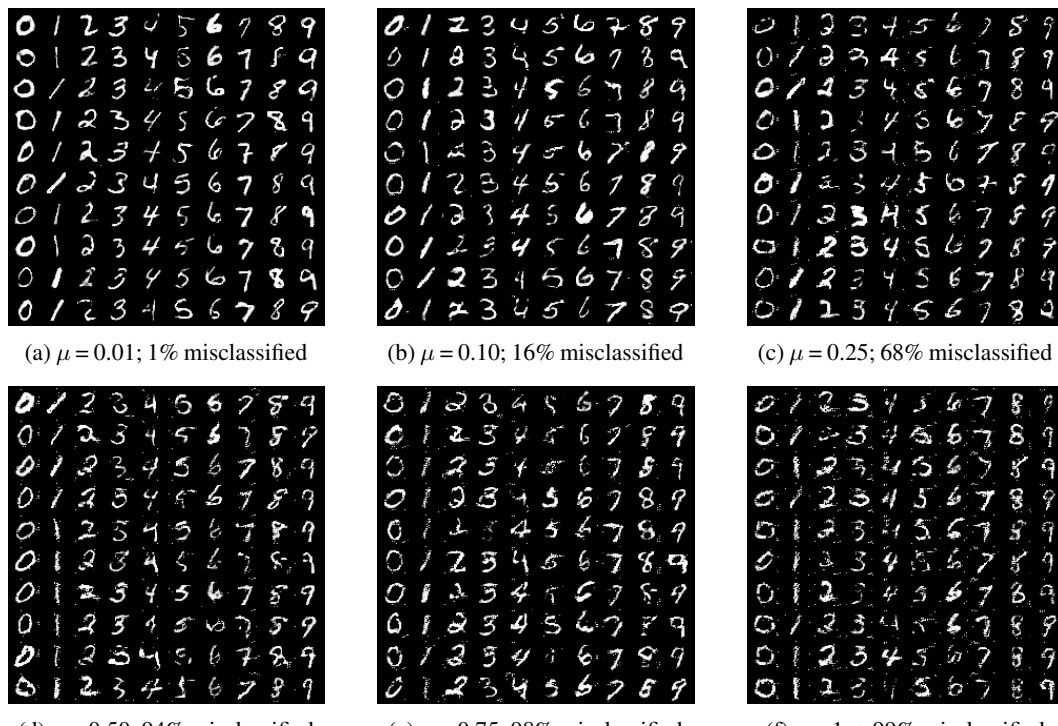

(a) $\mu = 0.01$; 1% misclassified    (b) $\mu = 0.10$; 16% misclassified    (c) $\mu = 0.25$; 68% misclassified

(d) $\mu = 0.50$; 94% misclassified    (e) $\mu = 0.75$; 98% misclassified    (f) $\mu = 1$; >99% misclassified

Figure 26: The effect of attack rate $\mu$ on image quality and proportion misclassified, using otherwise the same setup as in Figure 13. An attack rate of 1 is equivalent to not having the attack rate. As expected, lower attack rates mean higher visual quality, but a less successful attack.

