# OpenReview forum: "Adaptive Generation of Unrestricted Adversarial Inputs"
_ICLR.cc/2020/Conference — Reject_

### Official Review · AnonReviewer3 · 2019-10-22
**Official Blind Review #3**

**Rating:** 3

**Review:**

The paper proposes using GANs to generate unrestricted adversarial examples. They seek to generate examples that are adversarial for a specific classifier, and they do so by using class-conditional GANs and a fine-tuning loss. The fine-tuning loss consists of both the ordinary GAN loss (to fool the discriminator) as well as an adversarial loss (which rewards the GAN for generating examples misclassified by the specific classifier). The authors perform various experiments on their generated examples to check for realism and how adversarial the generated images are.

I would reject this paper for two key reasons. First, I feel that the contributions are not significant enough (in comparison to the prior work of Song et. al). Second, I feel that some of the methods (and some of the writing) are not too principled.

In my opinion, unrestricted adversarial examples are significant if they can be made to be realistic. If our current deep learning models often mislabeled very realistic images, that would properly expose a big failure mode of our current models. However, if our machine learning models perform poorly on images that look fake/generated 40% of the time (which is what the authors state) and don’t look too realistic to humans, it is less worrying.

In comparison to Song et. al, the authors state that their methods result in very similar results in terms of realism and how adversarial their images are (arguably, Song et. al actually produces better results in terms of being adversarial). In my opinion, the authors’ claimed improvements are not significant enough, because I think realism should be the primary metric to evaluate this field. Improving speed of generation is nice, and being able to bypass a simple adversarial training procedure is interesting but not significant unless this insight is expanded upon. The results on MNIST in Fig. 5 and Fig. 6 are not too convincing, as simpler attacks that generate (arguably) more realistic images like translations and rotations [1] or L1/L2 attacks [2] (since the networks are trained for L_inf robustness) can also degrade accuracy. Finally, I can also think of another reasonable baseline that I would have liked to see the authors compare their method against. Because the authors want to attack a specific network, they could have (1) generated realistic images using a pre-trained GAN (2) used a norm-bounded attack on the specific classifier and the generated GAN images. These images could be even more realistic if the norm-bound of the attack is fairly small, and would still be able to attack specific classifiers.

Finally, I am confused by the comparison to a not-fine-tuned GAN in Fig. 14/Fig. 15 and would appreciate a clarification so that I can understand the results. For example, what does it mean for intended true label = 9, target label = 0 to have 90% success in Fig. 15? Does this mean that when you try to generate a 9 with the GAN, the classifier misclassifies it as a 0 90% of the time? In particular, I’m struggling to understand what the target label is for the case of the not-fine-tuned GAN.

Secondly, I feel that there are many instances in the paper where the methods used are not explained in a principled way. For example, one of the key parts of this work is the fine-tuning loss function. Why does the loss function involve multiplying the ordinary GAN loss (with some additional transformation applied to it which seems unnecessary) with the adversarial loss? It seems most reasonable add the adversarial loss and the ordinary GAN loss (without the additional transformation). Is the stochastic loss selection procedure necessary? If all these peculiarities of the method are necessary, it seems that the success of this method is quite brittle.

Additional feedback:

- In the intro, I think citing [3] in addition to Xu et. al is more appropriate.
- You should refer to Figure 1 somewhere in the text of your work
- In section 3.2, you can use “cosine similarity” to describe what you are doing faster.
- When you talk about “global optima of realistic adversarial examples” and “local optimal of unrealistic adversarial examples,” it sounds weird. I would try to reword this because I don’t think you are trying to make a precise mathematical statement but it sounds like one when you write it this way.
- In Table 1, I would format the numbers better to be vertically aligned
- You should provide a citation for MixTrain on page 5

[1] https://arxiv.org/abs/1712.02779,
[2] https://arxiv.org/abs/1905.01034
[3] https://arxiv.org/abs/1802.00420

**Experience Assessment:**

I have read many papers in this area.

**Review Assessment: Checking Correctness Of Derivations And Theory:**

N/A

**Review Assessment: Checking Correctness Of Experiments:**

I assessed the sensibility of the experiments.

**Review Assessment: Thoroughness In Paper Reading:**

I read the paper at least twice and used my best judgement in assessing the paper.

---

> ### Author Response · Authors · 2019-11-12
> **Response to Reviewer #3 (Part 1)**
>
> Thank you for your detailed and thoughtful review. Although we are disappointed about your recommendation, we are grateful for the specificity and cogency of your feedback, which makes it especially easy to either improve our paper or respond to any points of disagreement.
>
> As we understand it, you have raised two specific concerns regarding the significance of our work: that unrestricted adversarial examples are significant only if they are realistic, and that our work is too incremental in comparison to the prior work of Song et al. We address these separately.
>
> Significance of Results: Realism of Unrestricted Adversarial Examples
>
> In criticising the realism of our generated adversarial examples, we believe you are directly addressing an important fundamental question: what is the purpose of adversarial examples research? Gilmer et al.’s seminal paper [1] on this subject identifies two motivations: security concerns, and improving understanding and capabilities of our models.
>
> Let us first consider security, for which it is essential to specify the precise threat model being used; Gilmer et al. suggest a compelling taxonomy of these. For ‘non-suspicious attacks’, the requirement is that the attack input must not be identifiable as being adversarial. Our experiments find that human judges are unable to identify which image is an unrestricted adversarial example 50% of the time on average (Fig. 5 in revised paper). Although this is not perfect (90%), this is still a far higher attack success rate than is desirable.
>
> However, this is not the only security threat model of interest. For Gilmer et al.’s ‘attacks with content constraints’ and ‘attacks without input constraints’ threat models, there is no requirement that the input be realistic, just that it fools the target system and maintains the correct semantics. All of the successful unrestricted adversarial examples in the paper would be a threat in such scenarios.
>
> But security implications are not the only motivation. Another framing of an ‘adversarial example’ is an input for which the network generalises in a different way to a human, therefore reaching the ‘incorrect’ answer. It would be desirable to have models that generalise correctly on all inputs, not merely on inputs indistinguishable from training data; unrestricted approaches such as ours need not have this property to identify interesting failures in generalisation which can then be studied by the community.
>
> In short, unrestricted adversarial examples need not be so realistic as to be indistinguishable from training data in order for them to represent a failure of generalisation which we would like to correct. Our contribution is not an improvement in realism or success rate, but a novel method with other advantages.
>
> Significance of Results: Comparison to Prior Work
>
> Our work is not incremental over Song et al. since it presents an entirely new method, with several important advantages.
>
> The most fundamental of these is that our method is adaptive. While $L_1$/$L_2$ attacks, rotation/translation attacks and Song et al.’s attack are all able to attack a network defended against $L_\infty$ attacks, they all share a weakness: they are not adaptive. That is, their attack procedure is fixed, and does not depend on the target network. It seems likely that all such attacks can be mitigated by adversarial training, since the classifier can learn not to rely on features targeted by that particular threat. Empirical studies show that this is true for $L_p$ perturbations [2], translations/rotations [3], and Song et al. (section 4.2).
>
> Conversely, our method is adaptive, since the generator is essentially finetuned to find a set of features to attack that the target classifier is reliant upon; this search is not constrained by an $L_p$ norm or any other requirement, and so the classifier is unable to anticipate all kinds of attack that the generator may learn next. Our preliminary attempts at defence against our attack have failed; we believe this challenge to be significant enough to be left as a fruitful direction for future work.
>
> As described in section 5.1, our method is also three orders of magnitude more efficient, demonstrably scales to a dataset orders of magnitude more complex than Song et al, and allows any existing GAN to be used out-of-the-box.
>
> [1] https://arxiv.org/abs/1807.06732
> [2] https://arxiv.org/abs/1908.08016
> [3] https://arxiv.org/pdf/1905.01034

---

> > ### Author Response · Authors · 2019-11-12
> > **Response to Reviewer #3 (Part 2)**
> >
> > Ablation Study for Section 3.3
> >
> > You correctly point out that we have not made clear how necessary each training strategy is. We have therefore carried out ablative experiments demonstrating the effect of omitting each in turn, reported in section 4.4 and appendix L. In short, pretraining and use of an attack rate other than 1 are not strictly necessary, but improve performance. Use of the naive loss function of simply summing the two loss terms degrades performance so badly as to be unusable. GANs are notoriously difficult to train at the best of times; adding an extra loss term which conflicts with the ordinary loss makes this even more difficult; strategies to improve training are helpful.
> >
> > Other Baseline
> >
> > Thank you for the suggestion to compare to norm-bounded perturbation attacks on images generated by a pretrained GAN. We have implemented this baseline, and found that the results are only slightly more effective than norm-bounded perturbations on the test set; this makes sense, since the pre-trained generator is supposed to have learnt this data distribution.
> >
> > Comparison to Non-Finetuned GAN
> >
> > We apologise for the confusion regarding the pretrained-only baseline - our description was unclear. To carry out the baseline, we first generate many examples with a particular intended true label, then filter these to keep only those which match the ‘target label’, then report the proportion of these are judged to indeed visually resemble the intended true label. We have updated the paper clarify this: please do let us know if we have still caused confusion.
> >
> > Additional Feedback
> >
> > The revision we have uploaded includes all of your suggestions of minor improvements to writing, referencing and formatting, for which we are grateful.
> >
> > We believe that this response fully addresses all the concerns you have raised - we look forward to hearing from you, either to raise your score or to continue the conversation with any further concerns you have.

---

> > > ### Comment · AnonReviewer3 · 2019-11-14
> > > **Thank you for your detailed response. Increasing score to 3 but still not in favor of acceptance.**
> > >
> > > Thank you for the detailed and clear response. I appreciate all the revisions the authors have made in response to my feedback, as well as the additional ablation studies and baselines that the authors ran in section 4.4. Thank you also for pointing out that GANS are notoriously tough to train, which clarifies why it is helpful to include all the modifications necessary to train the GAN. Overall, I think the paper has improved from before. Thus, I will increase my score from reject (1) to weak reject (3). However, I am still not convinced that the author's results are interesting enough for me to raise my score any further.
> > >
> > >
> > > I will explain my reasons for my current score of 3.
> > >
> > > 1) Regarding the realism of the generated images:
> > >
> > > I looked at a few BigGAN samples (which you fine-tune from) and looked at Imagenet samples from your appendix and it's fairly clear to me which images are more realistic. For example, I think very few of the Imagenet samples that you generate actually resemble anything realistic at all (e.g. your dog images are completely deformed and hardly have recognizable faces/eyes/noses/etc., while BigGAN dogs actually look fairly convincing). I am happy that you did realism tests with MTurkers on MNIST, but MNIST is a fairly simple dataset, and having 50% realism (where 90% is the best possible) on MNIST is not impressive to me either.
> > >
> > > More importantly, though, I disagree with the authors on one point. The authors argue that we want neural networks  to generalize to these particular unrestricted adversarial examples. For clarity, I am focusing on the Imagenet images the authors generate here. Because none of the authors' generated images really make much sense to a human (as a disclaimer, this is my personal opinion upon visual inspection of Figures 7,8,9,10 in the Appendix), why would we want or expect neural networks to produce anything sensible on these images? This is why I am stressing the realism so much.
> > >
> > >
> > > 2) Regarding the comparison to Song et. al:
> > >
> > > I do acknowledge that the authors have some improvements over Song et. al. Their method can be used with different GANs.
> > >
> > > I do not not think adaptivity is a particularly surprising or interesting result, as the authors generate unrestricted adversarial examples. I could easily do something close to an unrestricted gradient-based attack (for example, doing an L2 attack with a very very large epsilon), and this will probably generate some unrealistic image that fools a classifier. It's unreasonable (and not necessarily helpful) to require classifiers to be robust to all L2 attacks within a huge epsilon ball, just as it's unreasonable to expect classifiers to succeed on images that are frequently not realistic to humans.

---

> > > > ### Author Response · Authors · 2019-11-15
> > > > **Thank you - we believe we can address your further concerns (part 1)**
> > > >
> > > > Thank you for taking the time to receptively read and reflect upon our comments - we are of course very glad that you feel able to raise your score.
> > > >
> > > > We are also grateful that you have been so specific and clear about your reasons for not yet recommending acceptance, which once again makes it easy for us to improve our paper and respond to any points of disagreement. In particular, we believe we can allay your remaining concerns.
> > > >
> > > > ImageNet Realism: Lower-Quality BigGAN Checkpoint
> > > >
> > > > You are correct to point out that our ImageNet images are much less realistic than those published in the BigGAN paper. However, much of this is due to the underlying GAN that we are using. While we are using BigGAN, we can only use the publicly available code and checkpoints [1], which are not the same as those used to produce the images in the published BigGAN paper. In particular, there are two checkpoints available, both taken during a single training run: one after 100,000 iterations, and another taken later, “just before collapse”. This later checkpoint achieves an inception score of 97, considerably less than the score of 166.5 reported in the paper. However, we actually use the earlier checkpoint, with even worse performance, since the author recommends that it “may be easier to fine-tune”. Note that the resolution for these checkpoints is also 128x128, whereas the impressive images in the BigGAN paper are 512x512.
> > > >
> > > > ImageNet Realism: Hardware Limitations
> > > >
> > > > In addition, the codebase we use is intended for “4-8 GPUs” [1]. However, we only have access to one 16GB GPU. This means that the 15 is the greatest minibatch size we can use without running out of memory. This causes problems, since “a small batch leads to inaccurate estimation of the batch statistics, and reducing batch normalisation’s batch size increases the model error dramatically” [2]. This has caused problems for others attempting to use the same codebase [3], and the author has warned that using a smaller batch size “will likely negatively impact model performance” [4] and for this reason “this is not really a model for small hardware” [5]. Unfortunately, this is true for all state-of-the-art ImageNet GANs.
> > > >
> > > > In an attempt to clarify to what extent the unrealistic ImageNet results are a product of adversarial finetuning, rather than these external limitations, we have added images to the paper generated by the BigGAN on our machine after training (not adversarial finetuning) for 10 gradient steps. These can now be found in Appendix A. To our eyes, it appears that these images are little better than our adversarial examples, and so adversarial finetuning is not the primary cause of the unrealistic images (such as deformed dogs).
> > > >
> > > > MNIST Realism: Interpretation of Results
> > > >
> > > > You correctly point out that MNIST is a simple dataset. However, this in fact makes our results more impressive, not less impressive. The simplicity of classification means that the adversarially-robust classifiers are much better than for any other dataset, so finding adversarial examples is more challenging. Equally importantly, the highly-structured images (black background with simple white figures) makes it relatively easy to spot deviations from the usual data distribution. To be clear, our 50% figure is not that 50% of the time, human judges do not think the image looks realistic, but rather that 50% of the time, human judges are unable to distinguish our adversarial examples from examples in the dataset. This is not a trivial achievement.
> > > >
> > > > Our key message regarding realism is that unrestricted adversarial examples are valuable not only when they are indistinguishable from real data (for us, 50% of the time on MNIST), but also when they are unambiguous inputs for which a human could give a meaningful answer (as you point out). We achieve this on MNIST, and although you are correct to identify that GAN training issues hinder this on ImageNet, we still feel that these results are a significant step in a useful direction.
> > > >
> > > > [1] https://github.com/ajbrock/BigGAN-PyTorch
> > > > [2] https://arxiv.org/abs/1803.08494
> > > > [3] https://github.com/ajbrock/BigGAN-PyTorch/issues/40
> > > > [4] https://github.com/ajbrock/BigGAN-PyTorch/issues/39
> > > > [5] https://github.com/ajbrock/BigGAN-PyTorch/issues/31

---

> > > > > ### Author Response · Authors · 2019-11-15
> > > > > **(Part 2)**
> > > > >
> > > > > Adaptivity is Significant
> > > > >
> > > > > If we interpret your comment correctly, you state that adaptivity is a natural consequence of generating unrestricted adversarial examples. This is not the case: it is possible to use a fixed, easy-to-mitigate attack which is not constrained to norm-balls around test points. (Note that ‘unrestricted’ currently means just ‘not restricted to an $L_p$ norm ball’.) Our experiments show that Song et al.’s method is such an easy-to-mitigate method, and that ours is not.
> > > > >
> > > > > Another perspective on this is that all adversarial example algorithms so far involve a certain search procedure in image space for an example that fools the classifier, whereas our approach entails an optimisation over the weights of a generator, in effect searching for an adversarial example generation procedure which is effective against the target network.
> > > > >
> > > > > Although there is room for improvement regarding the realism of the generated images, it seems likely to us that further tuning and development of the procedure - and possibly scaling up the compute used - will remedy this. The tremendous pace of improvement in GAN image quality since 2014 is strong evidence in favour of this hypothesis.
> > > > >
> > > > > In short, the significance of introducing the first adaptive method for unrestricted adversarial example generation outweighs any current minor realism limitations imposed by GAN training difficulties.

---

> > > > > > ### Comment · AnonReviewer3 · 2019-11-15
> > > > > > **I have read the authors reply and will keep my score at 3.**
> > > > > >
> > > > > > Thank you for the clarifications. After reading your response, I still feel the same way as before regarding the significance of the proposed method, but it does clarify things further. I understand the authors had technical limitations, but I am still not convinced that the authors’ findings of such unrestricted adversarial examples is very useful.
> > > > > >
> > > > > > Again, I do appreciate the time the authors put into answering my questions.

---

### Official Review · AnonReviewer2 · 2019-10-22
**Official Blind Review #2**

**Rating:** 6

**Review:**

This paper presents a GAN architecture that generates realistic adversarial
inputs that fool a targeted classifier.  Adversarial inputs are unrestricted:
they may be any realistic images that humans will often classify as real
examples of the intended class, whereas the target model misclassifies them.
The novelty is that they finetune the generator itself during training, the
method can be applied to a variety of GAN architectures, and the method is fast.

Tricks used to successfully train the GAN are clearly described, and the
experimental evaluation was of good scope, covering a good selection of
experiments.  I particularly enjoyed the short Section 4.2 and Fig 7a+b, where
they show that a local defense can always be fooled somewhere else along
the input manifold of that class.

While the modifications to existing solutions may at first seem minor, they
have significant impact in applicability, effectiveness and speed of generating
unrestricted adversarial images.  So I think this paper can be accepted.

I had a bit of avoidable confusion in the introductory sections.  Figure 1
describing the GAN is never referred to.  It includes components not exactly
agreeing with my naive expectations from surrounding text.  Are any Fig. 1 features
optional?  It would help to highlight the novel elements in Fig. 1.  Or does
Fig.1 correspond perhaps to the combined GAN elements in Section 4 ("In our
experiments, we combine three ...").  My uncertainty was really relieved only
by the time I got to Related Work and Appendix E :(

The main claims seemed well supported by experiments, apart from claim 3
(applicability to "any" checkpointed GAN codebase). Might the scope of their
approach also be clarified by clearly identifying required and optional GAN
components in Fig. 1?

---- misc comments ----
Some sentences were long and difficult to parse:
- 4.1: "Our method generates..., else ...."  Perhaps make the else clause a second sentence.
- 4.2: "Image quality as measured ...." length and references made this difficult to read. Can you
rewrite as separate shorter sentences?


**Experience Assessment:**

I have read many papers in this area.

**Review Assessment: Checking Correctness Of Derivations And Theory:**

I assessed the sensibility of the derivations and theory.

**Review Assessment: Checking Correctness Of Experiments:**

I assessed the sensibility of the experiments.

**Review Assessment: Thoroughness In Paper Reading:**

I read the paper thoroughly.

---

> ### Author Response · Authors · 2019-11-12
> **Response to Reviewer #2**
>
> Thank you for your detailed and thoughtful review. We are especially glad to read that you find the experimental evaluation extensive and particularly enjoyed the demonstration that our method is able to find new ways of fooling standard adversarial training (which is very effective at mitigating the state of the art).
>
> We are sorry to hear that some parts of our exposition - especially Figure 1, which was indeed out-of-date - caused you (and another reviewer) some confusion. We have removed Figure 1 and rewritten our exposition in light of your feedback, and hope that this revision greatly improves the clarity of this point.
>
> To address your concern directly, any conditional GAN can be used with our method, with no other restrictions on the architecture. Although it is completely standard for a contemporary GAN to be class-conditional, we have clarified this condition in our revision.
>
> We have also incorporated your minor writing improvements, for which we are grateful.
>
> We believe that this response fully addresses all the concerns you have raised - we look forward to hearing from you, either to raise your score or to continue the conversation with any further concerns you have.

---

### Official Review · AnonReviewer1 · 2019-10-22
**Official Blind Review #1**

**Rating:** 6

**Review:**

======== update ========
I have read the authors' response and it has addressed most of my concerns. I am glad to see the authors' experiments on online adversarial training.

However, there is one additional concern that I didn't realize previously. Currently the performance of adversarial training is measured in "success rates". However it seems to me this success rates were not computed using human evaluation (since the authors claim once the classifier is finished training, the attack success rate can be larger than 99%). I would have changed my score to 8 if either 1) some adversarial images from the generator were included after finishing adversarial training or 2) success rates using human evaluation is reported. Unfortunately, I only realized this after the author rebuttal period, and the authors didn't have the chance to address this.

That being said, I feel this paper still presents interesting contribution to the field. I am still largely in favor of the acceptance of this paper, and will remain my rating of 6 for now. If this paper gets accepted, I strongly encourage the authors to address the concern I mentioned above in their camera ready.


======== original reviews ========

This paper proposes a novel method on generating unrestricted adversarial examples by finetuning GANs. The authors have conducted comprehensive experiments on evaluating the advantages of their approach. They demonstrated that their attack is harder to mitigate using adversarial training, produces unrestricted adversarial examples faster than existing methods, and can generate some unrestricted adversarial examples for complex high-dimensional datasets such as ImageNet.

I feel although the approach is straightforward, the authors have done a good job in motivating the method and have demonstrated its advantages via a good cohort of experiments. I like how the authors motivated finetuning in section 3.2, and I am glad that the authors have conducted ablative experiments to support their arguments in section 4.4. The experiments on adversarial training are especially interesting, since previous work hasn't considered this straightforward defense against unrestricted adversarial attacks. I am also glad that the authors can generate unrestricted adversarial examples for data as complicated as ImageNet images using the latent technique in GANs. Although still not perfect, some of the unrestricted adversarial examples on ImageNet are surprisingly good to the sense that they may be used as practical attacks.

The writing is great, and it is a pleasure to read this paper.

I do have some suggestions and questions for further improvement of the paper, and I strongly recommend the authors to address those before publication.

- Section 3 is lacking an explicit form of the combined objective function. Currently some loss functions such as $l_ordinary$, $l_targeted$, $l_d$ and $l_finetune$ are only defined in Figure 1 but not in the main text. It is not clear their explicit mathematical form.

- In section 3.2, it is better to also mention the ablative study you did later in section 4.4.

- In section 4.1, the authors showed nearest neighbors to some of the unrestricted adversarial examples they generated. It is more convincing to have some quantitive results of that. For example, what is the average minimum distance to training data for a group of 10000 unrestricted adversarial examples? In addition, what is the distance function used in computing nearest neighbors? Did you use Euclidean distance? If so, it would be better to also have results using distances computed in the feature space of a pre-trained convolutional network.

- In section 4.2, the adversarial training was done by alternating two phases of training rounds. I am wondering whether this makes the classifier harder to adapt to the newly generated unrestricted adversarial examples? Can you use some procedure more similar to traditional adversarial training, i.e., the attacker and the classifier are learned together at each step?

- Song et al. require 100-500 iterations to generate an adversarial example, whereas your approach only need one iteration. Why is your approach 400 to 2000x more efficient? What is the additional reason that speeds up your approach?

- In section 4.5 line 1, the word "replies" was repeated twice.





**Experience Assessment:**

I have published one or two papers in this area.

**Review Assessment: Checking Correctness Of Derivations And Theory:**

N/A

**Review Assessment: Checking Correctness Of Experiments:**

I assessed the sensibility of the experiments.

**Review Assessment: Thoroughness In Paper Reading:**

I read the paper at least twice and used my best judgement in assessing the paper.

---

> ### Author Response · Authors · 2019-11-12
> **Response to Reviewer #1**
>
> Thank you for your detailed and thoughtful review. We are especially glad to read that you find the method to be well-motivated, the empirical evaluation to be comprehensive, and the writing to be lucid.
>
> We are grateful for your suggestions of minor improvements to the clarity of the paper. We have uploaded a revision of the paper with these changes incorporated, including clarification of the overall objective function.
>
> You are correct to point out that Song et al. require 100-500 iterations to generate an adversarial example, yet we claim a 400-2000x efficiency improvement. The extra factor of four is because our method requires only a single forward pass through the generator network, while each iteration of Song et al. requires both forward and backward passes through both the generator and classifier networks. We hope that our revised wording makes this more explicit.
>
> While it could be interesting to repeat the nearest-neighbour calculations with a larger sample size, ten handpicked images are sufficient for a sanity check that our examples really are unrestricted. We believe a sanity check is all that is required: the only case in which our generated adversarial examples would be within a typical $L_p$ norm radius is if the generator were simply memorising dataset images. As a result we have prioritised other experiments and improvements during this time-limited response period.
>
> Your proposal of an adversarial training procedure in which the classifier is trained simultaneously online with the GAN finetuning is very sensible. We have implemented this experiment (see section 4.2), and found that our method is again able to easily evade this adversarial training procedure; the generator is always able to find a new way of fooling the classifier, since it has no restrictions.
>
> We believe that this response fully addresses all the concerns you have raised - we look forward to hearing from you, either to raise your score or to continue the conversation with any further concerns you have.

---

### Author Response · Authors · 2019-11-12
**Summary of Improvements Made**

Thank you to all three reviewers for your thoughtful and constructive comments. We have responded in detail to each point made, and have uploaded an updated version of our paper incorporating your feedback. The key changes that have been made are:
- Removal of out-of-date and confusing explanatory diagram identified by reviewers #1 and #2; redrafting of exposition in section 3.1 which we hope is a much clearer explanation of our method.
- Addition of additional adversarial training experiment (4.2) suggested by reviewer #1.
- Addition of further ablative experiments (4.4) for strategies outlined in section 3.3 to address concerns of reviewer #3.
- Addition of baseline (4.4) suggested by reviewer #3.
- Smaller corrections and writing improvements.

We hope that this improved paper together with our responses to your individual comments will reassure you and allow you to increase your scores.

---

### Decision · Program_Chairs · 2019-12-19

**Decision:**

Reject

**Comment:**

This paper presents an interesting method for creating adversarial examples using a GAN.  Reviewers are concerned that ImageNet Results, while successfully evading a classifier, do not appear to be natural images.  Furthermore, the attacks are demonstrated on fairly weak baseline classifiers that are known to be easily broken.  They attack Resnet50 (without adv training), for which Lp-bounded attacks empirically seem to produce more convincing images.  For MNIST, they attack Wong and Kolter’s "certifiable" defense, which is empirically much weaker than an adversarially trained network, and also weaker than more recent certifiable baselines.